# Effective Asthma Management: Is It Time to Let the AIR out of SABA?

**DOI:** 10.3390/jcm9040921

**Published:** 2020-03-27

**Authors:** Alan Kaplan, Patrick D. Mitchell, Andrew J. Cave, Remi Gagnon, Vanessa Foran, Anne K. Ellis

**Affiliations:** 1Family Physician Airways Group of Canada, Edmonton, AB T5X 4P8, Canada; 2Cumming School of Medicine, University of Calgary, Calgary, AB T2N 1N4, Canada; Patrick.Mitchell2@albertahealthservices.ca; 3Department of Family Medicine, Faculty of Medicine and Dentistry, University of Alberta, Edmonton, AB T6G 2R7, Canada; acave@ualberta.ca; 4Association of Allergists and Immunologists of Québec, Montréal, QC H5B 1G8, Canada; rgagnon@csacqc.ca; 5Asthma Canada, Toronto, ON M4S 2Z2, Canada; vanessa.foran@asthma.ca; 6Division of Allergy & Immunology, Department of Medicine, Queen’s University, Kingston, ON K7L 3N6, Canada; Anne.Ellis@kingstonhsc.ca

**Keywords:** SABA overuse, systemic steroid overuse, asthma control, mild asthma, ICS adherence, treatment, exacerbation

## Abstract

For years, standard asthma treatment has included short acting beta agonists (SABA), including as monotherapy in patients with mild asthma symptoms. In the Global Initiative for Asthma 2019 strategy for the management of asthma, the authors recommended a significant departure from the traditional treatments. Short acting beta agonists (SABAs) are no longer recommended as the preferred reliever for patients when they are symptomatic and should not be used at all as monotherapy because of significant safety concerns and poor outcomes. Instead, the more appropriate course is the use of a combined inhaled corticosteroid–fast acting beta agonist as a reliever. This paper discusses the issues associated with the use of SABA, the reasons that patients over-use SABA, difficulties that can be expected in overcoming SABA over-reliance in patients, and our evolving understanding of the use of “anti-inflammatory relievers” in our patients with asthma.

## 1. Introduction

In 2019, the Global Initiate for Asthma (GINA) changed their strategy for the management of step 1 and step 2 (mild) asthma: Short acting beta agonists (SABAs) as a monotherapy should not be used in patients with mild asthma; rather, patients should be prescribed a combination inhaled corticoid steroid (ICS)–fast acting reliever instead [1]. This change was made because of the recognition that SABA overuse and subsequent ICS underuse are responsible for safety concerns and poor outcomes, including hospitalization and possibly death. This change mostly impacts primary care physicians as they are most frequently providing care for these patients with mild asthma.

### The Problem

For many years, the standard asthma treatment has included the use of SABAs, either alone or as part of a therapy including inhaled corticoid steroids (ICSs), to provide rapid relief of symptoms in patients at the time when they are symptomatic, with the administration of oral corticosteroids (OCSs) to manage patients during moderate or severe acute exacerbations. However, with recent evolution in our understanding of the impacts of this reliance of SABA use on patient outcomes, the central role that these agents play in the treatment of asthma has been drawn into question.

While the use of as needed (PRN) SABA provides rapid relief of asthma symptoms, it does not address the underlying inflammatory process and does not protect the patient from exacerbations [2]. Patients that are treated with SABA alone are at higher risk for asthma related death [3] and urgent asthma-related healthcare [4] even if they have good symptom control [5]. Asthma patients treated with SABA monotherapy also have worse long-term outcomes and lower lung function than patients that are treated with low dose maintenance ICSs from the time of diagnosis [6]. Differences between patient and physician views of control and poor communication between patients and physicians about asthma status have been cited as a paradox of asthma management contributing to over-reliance on SABAs and underuse of ICSs [7].

All patients with asthma are at risk of exacerbation, regardless of the severity of the underlying disease [1]. Risk factors or indicators for exacerbation include high SABA use, environmental exposures (smoke, air pollution, or allergens), history of exacerbation, and poor adherence to therapy [1]. The rate of exacerbation is not trivial, even for patients with mild asthma. Clinical trial data show that patients with mild asthma treated with PRN SABAs still have a greater than 20% risk of a moderate to severe exacerbation over the course of one year [8]. In 2011, asthma was the leading cause for hospitalization in Canada, with emergency rooms dealing with more than 64,000 asthma related events [9]. Almost one third of asthma patients have reported having at least one emergency department visit each year [10]. The costs to society are profound; estimates of the direct costs to the health system (including hospitals, drugs, and physician fees) are $1.35 billion for 2020 with indirect costs (long term disability and mortality) adding a further $1.7 billion, for a total of over $3 billion [11]. Costs are expected to rise to $4.2 billion in the next decade [11].

Patients who are treated with SABA monotherapy have a greater risk of severe exacerbation than patients treated with ICSs; these exacerbations are routinely managed through the use of OCSs, typically in short courses or bursts [12]. However, there are very real concerns associated with the prescription of OCSs, particularly with adverse effects of these agents. These effects include increased risk of herpes zoster, cardiovascular events, type 2 diabetes, bone related conditions and fractures, cataracts, obesity, and hypertension [9,13]. The risk is strongly associated with cumulative OCS dose, rather than the maximum dose from a single course. Patients who receive even one to three prescriptions in a year have a 4% increased risk for adverse effects, while patients receiving four or more have a 29% increased risk [13]. However, the risk does not appear to return to baseline; patients who received four or more OCS prescriptions for any 3 years in the prior 10 had a 1.73-fold greater risk for adverse effects than patients who did not receive OCS prescription [13].

There are significant costs associated with OCS exposure to patients with asthma. Comparing patients that have been treated with OCSs to matched OCS untreated patients, long term adverse and costly outcomes have been associated with OCS initiation and have a dosage–response relationship with OCS exposure [14]. Voorham and colleagues found there were profound OCS dose-dependent health-care utilization increases in both general practitioner visits and prescriptions, as well as increases in costs of hospitalizations and general practitioner visits associated with OCS use. A recent Swedish observational study found that asthma patients who were regularly treated with OCSs had triple the health care utilization costs when compared with patients who were not treated with OCSs [15]. The major cost drivers were different in the two groups; the primary cost for OCS non-treated patients were primary care consultations, while the primary costs for OCS users were associated with inpatient care. More than 60% of the total costs for asthma and comorbidities was borne by patients regularly treated with OCSs, with more than 70% of the costs of each of asthma, osteoporosis, and pneumonia resulting from treatment of regular OCS users.

Table 1 illustrates the costs of OCS use in a UK cohort of asthma patients with a range of OCS dose levels [14]. The study noted that the negative impact of OCSs was dose related such that patients with higher calculated doses of ICSs experienced the greatest negative impacts for healthcare utilization, costs, and adverse effects [14].

## 2. Asthma Control

Guidelines recommend reviewing asthma control based on symptoms, in part on the amount of SABAs used per week. The current (2019) GINA strategy recommends SABA use < 3 times per week, with assessment for patients who consume three or more canisters per year (equivalent to 12 puffs per week) as there is an increased risk of asthma attack or exacerbation for patients using three or more canisters per year [1]. However, practicing doctors know that the patient reality is different. Patients value their rescue inhaler (SABA) and use it as needed, often underusing their controller medication (ICS and others).

Poor adherence is a common theme when treating chronic conditions and one that may seem to many health-care practitioners as particularly troubling with asthma. Indeed, poor adherence is common with adherence rates of around 50% in children and 30%–70% in adults (depending on the age, sex, ethnicity, and country) [16]. However, good adherence is associated with fewer exacerbations and better outcomes. The most common pattern is for patients to use medication when they have symptoms and avoid use when symptoms are not present. When symptoms occur, patients will increase their use of SABAs but not their controller medication [17]. Paradoxes of asthma treatment support patients’ poor understanding of disease control and support over-reliance on SABA medication [7]. Patients are initially treated with SABA monotherapy when they have infrequent symptoms, so they believe the symptom relief is the goal of therapy, rather than management of the underlying pathophysiology. They believe that they have the autonomy to make the decision when to use the inhaler. However, as the disease advances, patients lose the autonomy of the decision when to treat (with maintenance controller) and are told to avoid the medication that meets their goals for treatment (the reliever), leading to conflict in their minds [7]. A combination of reliever with anti-inflammatory would then meet both the goals of the clinician and the patient [18].

Practical issues with current guidelines for the treatment of asthma are experienced daily by both physician and patient and limit the effective treatment of this disease. Many of the real issues identified in current practice stem from the series of paradoxes that stem from guideline recommendations for asthma treatment [7,18]. These paradoxes include the following:conflicting messaging about the use of SABAs (encouraged in mild asthma but discouraged in more severe disease);assuming patient acceptance of advice to avoid use of SABAs, the medication that they perceive provides greatest benefit;different safety messages between SABAs and long acting beta-agonists (LABAs), where SABAs are considered safe;patient-physican dis-concordance between asthma control and frequency, impact, or severity of their symptoms;the patient’s perception of loss of autonomy over treatment when switching from a PRN SABA (in step one) to physician prescribed daily controller medication in step two [7].

Further, we now know that regular SABA use increases hyper-responsiveness in the lung, leading to greater sensitivity to triggers [18]. The use of a combination ICS with reliever (either SABA or fast acting LABA) across all patients has a better safety profile and efficacy in reducing exacerbation risk than a SABA alone [7,8,18,19].

## 3. How Did We Get Here?

With a greater understanding of the importance of reducing asthma exacerbations and the subsequent future risk of asthma-related mortalities, the recommendations for SABA use are being reconsidered. Specifically, for safety reasons, GINA no longer recommends SABA-only treatment in asthma but recommends that patients using three or more canisters of SABA per year should be assessed due to the associated increased future risk for exacerbations, hospitalization, or mortality [1]. A variety of factors increase the risk of exacerbations; patients may be at risk despite asthma severity, disease symptom control, or treatment adherence [20,21,22].

Historically we can look back to data from New Zealand where there was an epidemic of asthma deaths in the 1980s due to regular use of the extra-potent SABA fenoterol [23]. Further Canadian data from the early 1990s showed that there was a direct correlation between the number of SABA inhalers used and increasing mortality; for every one SABA inhaler increase, the death rate increased by a factor of two [24]. Several epidemiological studies have all reached the same conclusion: (over) reliance on SABA therapy is associated with increased adverse health outcomes including intubation and death [24,25].

Pathophysiologically, we now understand the anti-inflammatory benefit of inhaled corticosteroids (ICSs) on oedema and mucus over the sole acute bronchodilatory effect of SABAs. Previous consensus driven international strategies have recommended SABAs as first-line therapy to relieve immediate asthma symptoms (Figure 1A) [26]. This is despite the fact that asthma is most often characterized as an inflammatory condition characterized by exacerbations and persistent airflow limitations [7,18]. SABAs will resolve the immediate bronchospasm of an allergic trigger but have no inherent anti-inflammatory pharmacological properties and no effect on the late phase reaction. Further, regular use of SABAs have been shown to worsen the delayed inflammatory phase response [27], causing increased bronchial hyper-responsiveness on subsequent exposure to the trigger. Recurrent exposure of the airways to SABAs also contribute to a decreased response to SABA therapy as a reliever [28]. This could explain why regular use of SABAs (≥3 canisters of SABA/year) have an overall detrimental impact on the patient and have been linked to increased OCS use, increased emergency department visits, hospitalization, and disease progression [2,29].

The risk of SABA over-reliance has been clearly identified in a series of case reviews and studies. In a review of 192 patients in the UK who died from asthma in 2012, it was clear that there was a significant over-reliance on SABAs with 39% of those patients receiving 12 SABA inhalers per year, and 4% receiving 50 SABA inhalers in one year [30]! In contrast, the use of controllers was low, with 80% receiving less than the optimal 12 prescriptions per year and 38% receiving less than four per year. Another study of over 35,000 UK patients with asthma showed a direct relationship between the frequency of SABAs prescribed beyond the baseline of 1–3 per year and risk of hospital admission for asthma. The author’s conclusion was that there is a progressive risk of hospital admission associated with the prescription of 3 or more canisters of SABA per year [31].

A qualitative study [32] examined the reasons for SABA over-reliance in young adults with asthma. Reasons given for SABA inhaler over-reliance included wanting a short-term quick fix for asthma symptoms, cost, poor adaptation to illness, reducing stigma of having a chronic illness, asthma having career and social issues, fear of symptoms, anger at having illness, feeling that they cannot survive without it (the blue one), but that they could survive without the maintenance medicine. These issues require disease education and reinforcement of best practices.

It has been shown that shared decision making between the patient and clinician improves adherence to asthma therapy [33]. However, it did require four additional steps: i) identifying patient goals and preferences; ii) summarizing patient goals and preferences; iii) discussing relative merits of different treatment options in relation to goals and preferences; and iv) negotiating a decision about a treatment regimen. One study found that after one year, there was an increased total days’ controller prescription by an average of 77 days and by 9.6 ICS canister equivalents, increased quality of life by 0.4 points out of 5, reduced unscheduled asthma physician visits, reduced SABA acquisition by 1.6 canister equivalents, and doubling of the likelihood of having well controlled asthma [33].

## 4. Solutions

Data has shown that simultaneous anti-inflammatory and reliever use [8,34,35,36] demonstrates similar outcomes in asthma control with significantly reduced exacerbation rates over SABA alone. This has led to a paradigm shift which has been reflected in a change in the GINA strategy. As of 2019, based on strong evidence that SABA-only treatment increases the risk of severe exacerbations and asthma related death, SABA therapy should no longer be considered for monotherapy in asthma, with anti-inflammatory reliever therapy now the preferred reliever in adults (Figure 1B). Further, the recommendation in children is now the co-administration of ICSs with SABAs to ensure inflammation treatment is optimized at the time it is needed [1].

This change was based on significant evidence that SABA-only treatment increases the risk of severe exacerbations and asthma-related death and the supporting data that adding ICSs significantly reduces the risk. A large body of evidence from randomized controlled trials and observational studies shows that low dose ICSs substantially reduce the risk of severe exacerbations, hospitalizations, and death [3,4,37,38]. One study demonstrated that regular low dose ICS reduces the risk of asthma exacerbations by 60% and improves asthma control days by half [37]. Further, more recent data [8,34] has demonstrated that treatment with budesonide-formoterol (ICS-long-acting β-agonist) solely as-needed prevented exacerbations and loss of lung function, however, was less effective at mitigating symptoms than regular ICS maintenance therapy. Unfortunately, many patients do not adhere with regular ICS therapy and are not availing themselves of this protection. Patients require ICS at the time of worsening symptoms but have learned that SABAs provide the benefits of immediate relief that they seek [7]. Coupling the delivery of ICSs with a reliever makes use of the patient goals for therapy (i.e., fast relief) to ensure that the required anti-inflammatory is delivered when it is most needed, at the time of the asthma worsening [38].

The GINA strategy notes that patients receive either symptom-driven (in mild asthma) or daily ICS-containing controller treatment to reduce the risk of severe exacerbations and asthma-related death. Further recommendations are for as-needed controller treatment in mild asthma, with a preferred controller being low dose ICS–formoterol taken as needed for relief of symptoms and before exercise [1]. Currently, only budesonide–formoterol has been studied in Canada in this role; beclomethasone–formoterol has been approved in Europe for the treatment of moderate to severe asthma only. The use of SABA reliever in ICS controller therapy (either with or without a LABA) is associated with a higher risk of asthma exacerbation, compared to using budesonide–formoterol as maintenance and reliever, in patients over the age of 12 years [35]. At the time of this writing, only studies investigating budesonide–formoterol in this role have been conducted. Different ICS components may have different safety outcomes, and certainly different LABA compounds could lack the rapid onset required. While this concept may be applicable to other ICS–formoterol products, without controlled trials, recommending other ICS–formoterol products in this role would not be appropriate.

There are a number of questions that remain unanswered for many patients. Most trials of the anti-inflammatory reliever strategy have been performed using budesonide–formoterol in patients that are prescribed budesonide–formoterol as a controller (when they require a controller). However, in practice, some patients are using and are effectively managed with combination products other than budesonide–formoterol (such as fluticasone–salmeterol or mometasone–formoterol). The impact of budesonide–formoterol used as an as-needed reliever in the context of another controller (either a different ICS–LABA or just an ICS alone) has been poorly studied. In this case, patients who are well-controlled on these other controller medications should not be transitioned to the new strategy.

The continued use of a SABA monotherapy rather than an anti-inflammatory reliever may be appropriate in some patients. Patients with non-eosinophilic asthma (including neutrophilic asthma) and patients with true exercise induced bronchospasm (i.e., exercise induced asthma) may not benefit from the steroid component. Further, the Health Canada authorization for budesonide–formoterol indicates the combination as an anti-inflammatory reliever in patients with mild and persistent symptoms. SABAs should also be continued in patients that are using other ICS–LABA combinations as maintenance products; using budesonide–formoterol a reliever may not be appropriate in patients treated with a different ICS–LABA combination as these combinations have not been studied [39]. It is also important to ensure an accurate diagnosis of asthma in the patient; patients with mild COPD (or other ailments that may present as asthma) should not be treated with an ICS.

There may be some perceived disadvantages of anti-inflammatory reliever therapy. The side effects of the combination product are mild, well understood (as budesonide–formoterol is widely used as a maintenance therapy), and are generally related to the ICS component. The strategy encourages a lower total dose of the ICS as the inflammation is only treated at the time it is occurring, preventing overtreatment. Patients may be concerned with the cost of the product (when compared with short acting relievers). However, use of an anti-inflammatory reliever may ultimately reduce the quantity of the product used as well as reducing symptoms, partially negating the cost differential. There may be a concern with the total ICS dose in patients using an anti-inflammatory reliever strategy; however, the total dose for such a strategy will be less than patients using a SABA reliever in studies including mild, moderate, and severe asthma patients.

## 5. Conclusions

Despite the evidence, SABA over-reliance continues, with decreased ICS use due to poor adherence, increased OCS use due to exacerbations (typically more than 1 event per year for most patients), increased emergency department visits, and hospitalizations for our patients with asthma. While the negative effect of SABA over-reliance is understood, both clinicians and patients continue to reinforce this behaviour. We need to review SABA use with our patients, particularly to identify those using 3 or more SABA inhalers per year. In those patients, clinicians need to work with the patient to educate on the new GINA strategy of symptom-driven (mild asthma) or daily ICS-containing controller treatment, with as-needed controller treatment (low dose ICS–formoterol) taken as needed for relief of symptoms in place of SABA therapy. Data clearly shows that regular use of ICSs is the best strategy. However, lack of adherence to this often precludes use of ICSs. By taking advantage of the patient’s own behaviour to use an anti-inflammatory at the time of the reliever, we can impact behaviours and ensure an ICS is used at the time it is needed to control airway inflammation.

## Figures and Tables

**Figure 1 jcm-09-00921-f001:**
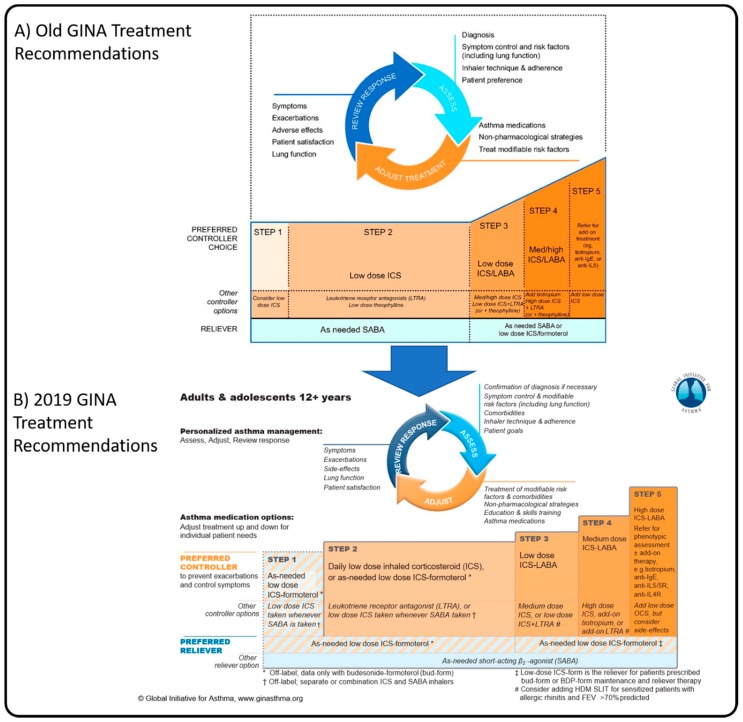
Personalized management for adults and adolescents from the Global Initiative for Asthma Global Strategy for Asthma Management and Prevention for 2018 (**A**) and 2019 (**B**), illustrating the changes to the recommended management in the newest strategy. Adapted from GINA 2018 [26] and GINA 2019 [1]. BDP: beclomethasone dipropionate; FEV: forced expiratory volume; HDM: house dust mite; LTRA: Leukotriene Receptor Antagonist; SLIT, sub-lingual therapy.

**Table 1 jcm-09-00921-t001:** Healthcare resource utilization and costs associated with oral corticosteroid (OCS) use in a matched historical cohort extracted from the UK Clinical Practice Research Datalink Database from 1994 to 2015.

**Annualized Health Care Utilization**		
**Factor**	**Non-OCS user IRR^1^** **(n = 9413)**	**OCS user IRR^1^** **(n = 9413)**
General Practitioner visits	1.00	1.22
Specialist visits	1.00	1.12
Hospitalization	1.00	1.14
Emergency Department Visits	1.00	1.26
Primary Care Prescriptions	1.00	1.35
**All Cause Health Care Costs**		
**Year**	**Non-OCS user Relative Cost**	**OCS User Relative Cost**
Year 1	100%	107%
Year 5	100%	150%
Year 10	100%	170%
Year 15	100%	210%
**15 Year Cumulative Incidence of Adverse Effects**		
**Adverse Effect**	**Non-OCS user Incidence** **(n = 9413)**	**OCS user Incidence** **(n = 9413)**
Renal Impairment	12.5%	27.9%
Type 2 Diabetes	5.6%	9.5%
Pneumonia	3.5%	11.3%
Cataracts	4.4%	11.0%
Cerebrovascular Event	5.1%	10.0%
Cardio-Cerebrovascular Disease	3.6%	9.9%
Osteoporosis	2.0%	8.0%
Myocardial Infarction	2.8%	7.3%
Heart failure	1.1%	3.6%
Glaucoma	1.7%	3.4%

^1^ IRR: incidence rate ratio.

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
