# Peer review of "Effective Asthma Management: Is It Time to Let the AIR out of SABA?"

_jcm, 2020, doi:10.3390/jcm9040921_

Round 1
Reviewer 1 Report
Kaplan and Colleagues discuss the issues associated with the use of SABA, which lead Global Initiative for Asthma 2019 to change strategy for the management of asthma. Authors correctly stress that all exacerbations are not associated with severe disease and that severe exacerbations in mild asthma represent a significant fraction of exacerbations requiring emergency consultation.
The paper is well written and clear. I think it is particularly useful for general practitioner, as many patients with mild disease, who continue to be treated with SABA monotherapy, have a greater risk of severe exacerbations, which are routinely managed through the use of short courses of oral corticosteroids.
Author Response
Thank you for your review of the manuscript.
The authors goal was to focus on primary care physicians as they are the healthcare professionals most closely invovled with the management of patients with mild asthma.
As no changes were requested, no changes were made due to this review.
Reviewer 2 Report
I read this article with great interest. This is an important issue, the change in our thinking about reliver in asthma treatment definitely require a sound and extensive analysis. I found the manuscript clear and well written.
I have small remarks, or rather questions that require answer in the manuscript.
I know that the financial support for preparation of this manuscript was provided by AstraZeneca Canada Inc. I also understand that only budesonide-formoterol has been studied in Canada in this role. On the other hand is there a need for further studies on other ICS in combination with formoterol? What are potential possibilities of future treatment based on the knowledge we have now? Would different ICS/formoterol registered as a reliver have any benefit for patients?
Another issue: in line 224 authors write "There are a number of questions that remain unanswered for many patients", but in the paragraph there is maybe one problem with continuation of treatment mentioned. What are indications for SABA alone? Should ICS-formoterol be use as a reliver in every case? Are there any important contraindications? Side effects to worry about? Benefits of less exacerbations were clearly mentioned. But are there any drawbacks?
Overall interesting and highly educational manuscript.
Author Response
Thank you for your review.
The authors agree that this area needs further research to gain a better understanding of the role of SABA and anti-inflammatory relievers in treatment of patients with mild asthma.
In response to the reviewer comment "On the other hand is there a need for further studies on other ICS in combination with formoterol? What are potential possibilities of future treatment based on the knowledge we have now? Would different ICS/formoterol registered as a reliver have any benefit for patients?"
- Beginning on Line 223 of the manuscript, we have added a short discussion on the current research conducted on anti-inflammatory reliever. While research into other formulations would be beneficial, in reality, conducting such research is prohibitively expensive and unlike to happen.
In response to "There are a number of questions that remain unanswered for many patients", but in the paragraph there is maybe one problem with continuation of treatment mentioned. What are indications for SABA alone? Should ICS-formoterol be use as a reliver in every case? Are there any important contraindications? Side effects to worry about? Benefits of less exacerbations were clearly mentioned. But are there any drawbacks?
- We have included, beginning at line 237, an expanded discussion on identifying appropriate patients, including current indications for budesonide-formoterol as anti-inflammatory reliever and ensuring accurate diagnosis of asthma. Further, we have included a discussion of the perceived disadvantages and concerns with the treatment strategy.